# Comparative Analysis of the Neutralizing Capacity of Monovalent and Bivalent Formulations of Betuvax-CoV-2, a Subunit Recombinant COVID-19 Vaccine, Against Various Strains of SARS-CoV-2

**DOI:** 10.3390/vaccines12101200

**Published:** 2024-10-21

**Authors:** Anna V. Vakhrusheva, Ekaterina A. Romanovskaya-Romanko, Marina A. Stukova, Maria M. Sukhova, Ksenia S. Kuznetsova, Aleksandr V. Kudriavtsev, Maria E. Frolova, Taras V. Ivanishin, Igor V. Krasilnikov, Artur A. Isaev

**Affiliations:** 1Betuvax, 119571 Moscow, Russia; suhova@betuvax.ru (M.M.S.); kuznecova@betuvax.ru (K.S.K.); ivanishin@betuvax.ru (T.V.I.); 2Department of Vaccinology, Smorodintsev Research Institute of Influenza, Ministry of Health of the Russian Federation, 197376 Saint Petersburg, Russia; romromka@yandex.ru (E.A.R.-R.); marina.stukova@influenza.spb.ru (M.A.S.); 3Research Institute for Systems Biology and Medicine, 117246 Moscow, Russia; akudriavtsev@sysbiomed.ru; 4Artgen Biotech, 119571 Moscow, Russia; frolova@nextgene.ru (M.E.F.); art.isaev@gmail.com (A.A.I.); 5Biotechnology Developments, 119571 Moscow, Russia; i.krasilnikov@rbiotech.ru

**Keywords:** SARS-CoV-2, COVID-19, subunit vaccines, bivalent vaccines

## Abstract

SARS-CoV-2, the causal agent of the COVID-19 pandemic, is characterized by rapid evolution, which poses a significant public health challenge. Effective vaccines that provide robust protection, elicit strong immune responses, exhibit favorable safety profiles, and enable cost-effective large-scale production are crucial. The RBD-Fc-based Betuvax-CoV-2 vaccine has previously demonstrated a favorable safety profile and induced a significant anti-SARS-CoV-2 humoral immune response in clinical trials. Due to the rapid evolution and emergence of new SARS-CoV-2 strains, the relevance of bivalent vaccine formulations has increased. Methods: This study compared the neutralizing capacity of monovalent and bivalent vaccine formulations against different SARS-CoV-2 strains detected with a SARS-CoV-2 microneutralization assay (MNT). Findings: The monovalent Wuhan-based vaccine generated neutralizing antibodies against the Wuhan and Omicron BA.2 variants but not the distinct Omicron BQ.1 strain. Conversely, the monovalent BA.2-based vaccine induced neutralizing antibodies against both Omicron strains but not Wuhan. While the bivalent Wuhan and BA.2-based vaccine was effective against strains containing the same antigens, it was insufficient to neutralize the distinctive BQ.1 strain at a small dosage. Interpretation: These findings suggest that the vaccine composition should closely match the circulating SARS-CoV-2 strain to elicit the optimal neutralizing antibody response and include the appropriate dosage. Moreover, this study did not find additional advantages of using the bivalent form over the monovalent form for the vaccination against a single prevailing SARS-CoV-2 strain.

## 1. Introduction

The COVID-19 pandemic, caused by the severe acute respiratory syndrome coronavirus 2 (SARS-CoV-2), was declared by the World Health Organization (WHO) on 11 March 2020. In a meeting on 5 May 2023, the WHO’s Emergency Committee announced that COVID-19 is now a well-established problem, no longer warranting a Public Health Emergency of International Concern. However, new SARS-CoV-2 strains continue to circulate and evolve globally, producing novel mutations and variants. Consequently, the specific timing, mutations, and antigenic characteristics of potential future variants, as well as the associated risks to public health, remain unpredictable.

As of 31 March 2024, the global COVID-19 pandemic has resulted in over 774 million confirmed cases and more than seven million reported deaths worldwide [1]. The WHO classifies variants of COVID-19 into several groups based on significance, dividing them into variants of interest (VOI) and variants under monitoring (VUM). At the moment, there are no SARS-CoV-2 variants that meet the criteria to be classified as Variants of Concern (VOC). The ongoing evolution of SARS-CoV-2 and the emergence of new variants have necessitated the continuous development and evaluation of vaccines capable of conferring broad and durable protection against the virus.

As of June 2024, the WHO has reported the prevalence of new Omicron sub-lineages, including XBB.1.5 (“Kraken” XBB.1 + S:S486P), XBB.1.16 (XBB.1 + S:E180V, S:478R, S:F486P), EG.5 (XBB.1.9.2 + S:F456L), BA.2.86, and JN.1 (BA.2.86 + S:L455S). The latter two, BA.2.86 and JN.1, are currently classified as Variants of Interest (VOIs).

The new BA.2.86 descendant, JN.1, features an additional L455S mutation in the Spike protein, which may contribute to its rapid growth. As of the epidemiological week of 18 to 24 March 2024, there are currently 162,773 JN.1 sequences available from 121 countries, representing 93.7% of the globally available sequences [2].

The BA.2 lineage, which gave rise to BA.2.86, is characterized by its numerous fifty-seven mutations and increased transmissibility. Compared to the BA.1 receptor-binding domain (RBD) of the Spike protein, the BA.2 RBD harbors four unique mutations—L371F, T376A, D405N, and R408S—that contribute to increased virus infectivity [3].

Rapid SARS-CoV-2 evolution, high immune evasion, and transmission capacity necessitate high vaccine coverage. Consequently, the trend in vaccine development has shifted towards bivalent vaccines. The ability to elicit broader and more durable immune responses against multiple SARS-CoV-2 variants is supposed to be a crucial advantage of bivalent formulations over monovalent ones [4].

Nevertheless, the superior performance of bivalent vaccines compared to monovalent vaccines remains uncertain. The latest WHO guidance indicates that leading pharmaceutical manufacturers are updating their vaccine antigens to the monovalent JN.1 lineage formulation [5]. Therefore, we aimed to conduct a comparative evaluation of the neutralizing capacity between bivalent and monovalent formulations of the recombinant subunit vaccine “Betuvax-CoV-2”.

The subunit vaccine Betuvax-CoV-2 consists of a recombinant RBD-SD21-SD14-Fc (RBD-Fc) fusion protein attached to 100–180 nm betulin-based spherical nanoparticles [6], designed to enhance vaccine immunogenicity by mimicking the virus size and morphology (Figure 1).

Moreover, two peptide epitopes were added to the RBD antigen: the S14P5 epitope, located in the SD1 subdomain behind the RBD, and it appears that antibodies to this region sterically prevent virus binding to the ACE2 receptor; and the S21P2 epitope, located in the S2 region overlapping with the fusion peptide, and antibodies to this region prevent the virus from fusing with the target cell. The order of the epitopes was changed to improve protein stability (Figure 1).

Previously, the Betuvax-CoV-2 vaccine containing the Wuhan antigen has demonstrated a high safety profile and elicited good immunogenicity, as evidenced by the titers of total IgG and neutralizing antibodies against the RBD of SARS-CoV-2 in non-clinical and clinical studies. The clinical trial is registered with ClinicalTrials.gov (Study Identifier: NCT05270954). In Phase 2, both evaluated dosing regimens of the Betuvax-CoV-2 vaccine, administered via intramuscular injections at 20 µg + 5 µg and 20 µg + 20 µg, were determined to be safe and well-received. The majority of adverse events were mild, with only two serious adverse events reported, unrelated to the investigational drug. The results indicated a significant and durable humoral immune response against SARS-CoV-2 following two doses of Betuvax-CoV-2. Notably, these antibody levels did not drop substantially even by days 91 and 181 after the initial vaccine dose [7,8].

Given the importance of booster vaccinations and the need for ongoing updates to vaccine antigens, a critical consideration for vaccines intended for repeated administration is to minimize the potential for adverse effects. Consequently, among the array of vaccine platforms available, recombinant subunit vaccines, including Betuvax-CoV-2, emerge as an attractive option due to their notable safety profile and reduced reactogenicity while still maintaining robust immune responses. Therefore, it is essential to explore the optimal formulation of these vaccines. This research study is centered on comparing the neutralization capacity of bivalent and monovalent versions of the Betuvax-CoV-2 vaccine.

## 2. Methods

### 2.1. Animals and Ethics Statement

This study used BALB/c mice as the experimental model, a commonly utilized choice in vaccine immunogenicity research due to their well-understood immune system, robust Th2 response, and genetic uniformity that ensures replicable results. Female BALB/c mice, aged between 6 and 8 weeks at the initiation of the primary immunization, were procured from the Stolbovaya branch of the Scientific Center of Biomedical Technologies of the Federal Medical and Biological Agency of the Russian Federation. The permissible deviation in body weight of the animals from the group mean did not exceed 20%.

The quantity of mice utilized in this study was adequate to evaluate the vaccine’s immunogenicity and adhere to ethical research standards. All animals were housed in standard conditions consistent with Directive 2010/63/EU of the European Parliament and the Council of the European Union dated 22 September 2010 concerning the welfare of animals used for scientific purposes. Animal care procedures were in accordance with the respective facilities’ Standard Operating Procedures (SOPs) and the sanitary and epidemiological regulations described in SR 3.3686-21, “Sanitary and Epidemiological Requirements for the Maintenance of Experimental Biomedical Facilities (Vivariums)”.

The mice were housed in autoclaved sterile polycarbonate cages filled with hypoallergenic hardwood (sawdust) bedding material. They were provided with standard food and clean water ad libitum and maintained at a temperature range of 18–22 °C, a relative humidity level of 55–65%, and a light/dark cycle of approximately 12 h each day.

All procedures involving animals in this study were reviewed and approved by the Ethics Committee of the Smorodintsev Research Institute of Influenza for compliance with regulatory acts (protocol No. 10, dated 30 March 2023).

### 2.2. Formulations and Dosages of the Test Drug

Monovalent Betuvax-CoV-2 is an injectable suspension intended for intramuscular use, containing 20 μg/0.5 mL (1 dose) of the recombinant receptor-binding protein of the SARS-CoV-2 virus (RBD-Fc). This includes 20 μg of the Wuhan strain or Omicron BA.2 strain, along with a corpuscular adjuvant made from natural betulin (200 μg) and tris(hydroxymethyl)aminomethane (Tris) (3 mg). Additional components include sodium chloride (4.4 mg) and water for injections (up to 0.5 mL) as excipients.

The bivalent Betuvax-CoV-2 is also an injectable suspension for intramuscular use, with 20 μg/0.5 mL (1 dose). It contains a total of 20 μg of the RBD-Fc derived from both the Wuhan strain (10 μg) and the Omicron BA.2 strain (10 μg). Similar to the monovalent version, the bivalent vaccine includes a corpuscular adjuvant with natural betulin (200 μg) and Tris (3 mg), as well as sodium chloride (4.4 mg) and water for injection (up to 0.5 mL) as excipients.

The placebo is an injectable suspension containing Tris–HCl 0.05 M and NaCl 0.15 M, designed for intramuscular administration, with a volume of 0.5 mL per dose.

The preparations are stored and transported at temperatures 4–8 degrees Celsius, with established stability confirmed to last for over one year.

### 2.3. Research Design

The characteristics of the study groups and the schedule of manipulations are presented in Table 1.

### 2.4. Immunization of Animals and Serum Collection

Intramuscular immunization of animals was carried out by injecting thigh muscles with sterile disposable syringes with a needle size of 29 G, containing volumes of 500 μL each. Blood collection techniques and sample volumes were consistently standardized.

Serum collection was conducted within the timelines specified in the study design. Blood was obtained from the submandibular vein, after which blood samples were stored at room temperature for 2 h, centrifuged at 5000 rpm for 10 min, and the resulting whole serum was collected. The serum was stored at −20 °C until use.

### 2.5. Evaluation of Humoral Immune Responses

At the time points specified in the study design, serum was collected from animals to assess systemic IgG antibodies specific to the RBD-Fc protein from various variants of the SARS-CoV-2 virus. This assessment was performed using an enzyme-linked immunosorbent assay (ELISA). The assessment of the content of virus-neutralizing antibodies against different variants of the SARS-CoV-2 coronavirus in animal blood sera was conducted using the microneutralization reaction (MNT).

#### 2.5.1. ELISA

For the analysis, 96-well immunological plates from Medisorp (Nunc) were used. Recombinant RBD-Fc protein of the SARS-CoV-2 virus (Wuhan variant, Omicron BA.2, or Omicron BQ1.1) at a concentration of 1 μg/mL was used as the antigen. Antigen adsorption was carried out overnight at 4 °C. Washing was performed with PBS-T buffer (0.1% Tween20 in PBS) using an ELx50 automatic washer (BioTek). Blocking was carried out with 5% dry milk (Stoing, Russia) for 2 h at room temperature. Subsequently, serial dilutions of sera (starting from 1:100/1:800 for vaccinated animals and 1:50 for animals from the control group) were added to the plates and incubated in the plate wells for 1.5 h at room temperature. Goat Anti-Mouse IgG (H+L) peroxidase conjugate (“Abcam” ab97040, dilution 1:7500) was used for IgG detection. Incubation with the conjugate was carried out for 1 h at room temperature, followed by the addition of TMB substrate (“Bioservice”) and incubation for 15 min in the dark. To stop the reaction, 1N H2SO4 (“VEKTON”) was added. Optical density (OD) was measured using the CLARIOstar microplate reader (“LABTECH”), and the TMB-absorbing optical density was calculated as the difference in OD_450nm_–OD_620nm_. In the absence of a specific signal in the well with a serum dilution of 1:800 for vaccinated animals and 1:50 for control animals, the IgG titer value was considered as 400 and 25, respectively.

#### 2.5.2. MNT

The following strains of the SARS-CoV-2 were used for the MNT: Wuhan—hCoV-19/Russia/SPE-RII-27029S/2021 strain, GISAID EPI_ISL_1257814, passage 3; BA.2—hCoV-19/Russia/SPE-RII-16180S/2022 strain, GISAID not uploaded, passage 2; and BQ.1—hCoV-19/Russia/SVE-RII-MH110277S/2022 strain, GISAID EPI_ISL_15920109, passage 2. Animal sera were heated for 60 min at 56 °C to inactivate the complement system. Then, in 96-well plates with U-shaped bottoms (“Medpolimer”), dilutions of sera in AlphaMEM medium (“BioloT”) with the addition of 1% antibiotic–antimycotic (“Gibco”) and 2% preheated FBS serum (“Gibco”, #10500) in a volume of 60 μL were prepared, starting from 1:10 (relative to whole serum). To the serum dilutions, 25–50 TCID50 of the SARS-CoV-2 were added in an equivalent volume and incubated for 1 h at 37 °C. Then, 100 μL of dilutions were transferred to Vero cells (Vero-CCL-81, ATCC) in 96-well plates. The plates were incubated for 4–5 days at 37 °C until the development of characteristic cytopathic effects (CPE). The neutralizing titer was considered the highest serum dilution at which complete inhibition of viral infection was observed (i.e., absence of CPE). If no inhibition of viral infection was observed at a serum dilution of 1:10, the titer value was considered as 5. Manipulations with live SARS-CoV-2 were carried out in a containment zone equipped for work with BSL II group pathogens, based on the Smorodintsev Research Institute of Influenza.

### 2.6. Primary Data and Statistical Analysis

Statistical analysis of primary data was performed using Microsoft Office Excel 2010 and GraphPad Prism v6.01 software packages. The following statistical indicators were used for data representation: geometric mean and standard deviation.

For repeated within-group measurements, two-way repeated measures analysis of variance (two-way ANOVA) with Sidak’s post-test was conducted. To determine the significance of differences between group means, one-way analysis of variance (ANOVA) for group comparison was used, while for its non-parametric counterpart, the Kruskal–Wallis test was used. Subsequently, in case the null hypothesis was rejected, the Tukey test for multiple intergroup comparisons or the Dunnett’s test and non-parametric Dunn’s test for post hoc pairwise comparisons with the control group were employed. The a priori level of significance was set at α = 0.05. Differences were considered significant at an achieved significance level of *p* < α.

## 3. Results

The comparative study of the immune response of the monovalent and bivalent forms of the Betuvax-CoV-2 vaccine was conducted in BALB/c mice. The immunogenicity assessment of the samples was conducted after a single and double vaccination of the animals, performed with a 28-day interval in-between (Figure 2). Three groups of animals were administered samples of the candidate vaccine intramuscularly (total amount of coronavirus antigen RBD-Fc 20 µg, betulin adjuvant 200 µg in 0.5 mL suspension for intramuscular administration), and one control group had received a placebo (Tris–HCl buffer 0.05 M, NaCl 0.15 M) intramuscularly.

The formation of the IgG-specific serum antibodies to the monovalent Wuhan/monovalent BA.2/bivalent Wuhan + BA.2 coronavirus component of the Betuvax-CoV-2 vaccine was assessed after single and double immunizations in an ELISA against the RBD of Spike protein (Figure 3). Titers of specific IgG antibodies in mice were compared between single and double immunizations, as well as within the placebo group.

This study shows that even a single vaccine administration led to the formation of serum antibodies specific to the RBD protein of all tested SARS-CoV-2 strains (Wuhan/BA.2/BQ.1), significantly different from the placebo group (*p* < 0.0001). Double immunization resulted in a significant increase in titers of serum antibodies to the coronavirus component for each of the vaccine preparations used. The magnitude of the increase in all intramuscularly administered groups was similar, ranging from 18 to 20 times for Wuhan RBD protein (Figure 3A), 14–18 times for BA.2 RBD protein (Figure 3B), and 9–18 times for BQ.1 RBD protein (Figure 3C) compared to titers after single vaccination. No significant differences in GMT values between groups of animals vaccinated intramuscularly with monovalent and bivalent Betuvax-CoV-2 formulations were observed (*p* > 0.999).

The formation of the neutralizing antibodies to the monovalent Wuhan/monovalent BA.2/bivalent Wuhan + BA.2 coronavirus component of the Betuvax-CoV-2 vaccine was assessed after double immunizations in an MNT assay against different strains of SARS-CoV-2 (Figure 4). The titers of neutralizing antibodies in mice were compared between double immunizations and the placebo group.

After a single immunization, the formation of neutralizing antibodies has not been demonstrated for any of the investigated groups against any of the used coronavirus variants. Double immunization of animals with vaccine preparations resulted in the appearance of neutralizing antibodies against coronavirus variants with varying intensity.

Upon intramuscular administration of vaccine preparations, the formation of virus-neutralizing antibodies was only demonstrated in groups where the vaccine included the Wuhan variant protein: “Wuhan 20 μg” and “Wuhan 10 μg + BA.2 10 μg”. In animals vaccinated with a vaccine containing the RBD-Fc protein of the Omicron BA.2 variant, virus-neutralizing antibodies against the SARS-CoV-2 Wuhan strain were not formed. This indicates the inability of the RBD-Fc protein of Omicron BA.2 to generate antibodies with cross-neutralizing activity against the genetically earlier Wuhan variant of the coronavirus. The GMT values in the “Wuhan 20 μg” and “Wuhan 10 μg + BA.2 10 μg” groups were similar; however, the “Wuhan 10 μg + BA.2 10 μg” group showed a greater dispersion of individual values (Figure 4A).

To test the hypothesis of no differences in neutralizing antibody titers after double vaccination between groups, a statistical one-way analysis of variance (ANOVA) was conducted, followed by a pairwise comparison of experimental groups with their respective placebo control groups using the Dunnett criterion. Upon intramuscular double administration of vaccine preparations, it was shown that GMT values of neutralizing antibodies in the “Wuhan 20 μg” and “Wuhan 10 μg + BA.2 10 μg” groups were significantly higher compared to the placebo group (*p* < 0.0001).

The study findings show that double intramuscular immunization with a vaccine containing 20 μg of the RBD-Fc protein from the BA.2 variant elicited the highest titers of neutralizing antibodies against the Omicron BA.2 coronavirus strain (“BA.2 20 μg”). In contrast, the “Wuhan 10 μg + BA.2 10 μg” group receiving a bivalent vaccine with 10 μg each of the Wuhan and BA.2 antigens displayed lower neutralizing antibody levels, approximately 2.2 times less than the “BA.2 20 μg” group. The “Wuhan 20 μg” group exhibited the lowest neutralizing antibody response, indicating limited cross-reactivity of the Wuhan antigen against the BA.2 strain of SARS-CoV-2 (Figure 4B). The results demonstrated that double intramuscular administration of the vaccine formulations led to significantly elevated GMT of neutralizing antibodies against the Omicron BA.2 variant across all treatment groups in comparison to the placebo group (*p* < 0.0001, *p* < 0.01).

The analysis revealed that compared to the Wuhan and BA.2 coronavirus strains, the Omicron BQ.1 variant elicited a less robust immune response during the vaccination with distinctive forms of antigens, namely Wuhan and Omicron BA.2. Among the three experimental groups, neutralizing antibodies against BQ.1 were generated in the group receiving the “BA.2 20 μg” RBD-Fc protein as the vaccine component. The “BA.2 20 μg” group had a GMT of 17.8, while the “Wuhan 10 μg + BA.2 10 μg” group, with half the BA.2 antigen concentration, had a GMT of 10.0, and the “Wuhan 20 μg” group exhibited the lowest GMT of 6.3 (Figure 4C).

This is likely due to some evolutionary proximity between the Omicron BA.2 and Omicron BQ.1 variants, in contrast to the more distant Wuhan strain. These findings suggest that the formation of neutralizing antibodies against the BQ.1 variant is primarily driven by the amount of BA.2 antigen in the bivalent vaccine, while the contribution of the Wuhan RBD-Fc protein is limited. The statistical significance of the differences was demonstrated only for the monovalent “BA.2 20 μg” group (*p* = 0.0378).

## 4. Discussion

Recent research suggests that both bivalent and monovalent variant vaccine boosters can improve protection against known and emerging SARS-CoV-2 antigenic variants [9]. Some studies indicate that highly effective vaccination with broad coverage can be achieved through formulations that contain antigens matching both the original SARS-CoV-2 strain and the predominant variants of concern [10,11].

Different components of updated bivalent COVID-19 vaccines have been investigated, including the original SARS-CoV-2 strain and variants such as D614G, Alpha, Beta, and Omicron sub-lineages BA.1, BA.4, and BA.5 [12,13,14]. However, the rapid spread of the mutating Omicron variant prompted the accelerated emergency use authorization of bivalent vaccines targeting Omicron. In the United States, an mRNA vaccine combining the ancestral strain and the BA.4/BA.5 sub-lineages was approved on 31 August 2022. The systematic review spanned across eight publication databases from 1 September 2022 to 8 November 2023, in order to assess the absolute and relative effectiveness of bivalent mRNA vaccines targeting SARS-CoV-2 BA.1 and BA.4/5 variants. This analysis comprised 28 studies involving 55,393,303 individuals. The absolute vaccine effectiveness against symptomatic infection was recorded at 53.5% when compared to unvaccinated individuals, while the relative effectiveness stood at 30.8% and 28.4% in comparison to those who had received ≥2 and ≥3 original monovalent doses, respectively. Moreover, the bivalent formulations exhibited greater pooled effectiveness against severe outcomes, surpassing rates of 70.0% when compared to unvaccinated individuals [15]. These results highlight the superior protective advantages of bivalent boosters, particularly in preventing severe events, emphasizing the critical need to enhance vaccine coverage, particularly among vulnerable older populations [16].

Nevertheless, other studies comparing the vaccine effectiveness of an mRNA-based bivalent booster to a monovalent booster suggest that the bivalent vaccines have a less significant impact than their monovalent counterparts [17]. In a separate study, the bivalent vaccine showed an additional 8% protection against symptomatic Omicron BA.5 infection compared to the monovalent vaccines [18]. A systematic review revealed that both bivalent and monovalent boosters triggered strong immune responses, with bivalent vaccines generating similar antibody responses against both original and new variant strains [19]. However, the beneficial difference in the bivalents was not vast, potentially due to “imprinting” effects from prior exposures [20].

In this study, we also found that intramuscular administration of the vaccine preparations led to similar levels of RBD-specific IgG antibodies across the groups, regardless of the monovalent or bivalent formulation. A single dose significantly increased antibody titers compared to placebo, and a second dose further boosted titers for all variants tested, without significant differences between groups.

However, the vaccine formulations differed in their ability to elicit neutralizing antibodies. For the Wuhan strain, neutralizing antibodies were only generated in the Wuhan RBD-Fc groups, not the BA.2 RBD-Fc group. For Omicron BA.2, all groups showed higher neutralizing antibody levels than placebo, with the 20 μg BA.2 RBD-Fc group having the highest titers. For Omicron BQ.1, a less pronounced neutralizing response was seen, with the BA.2 RBD-Fc monovalent form showing the most robust response compared to the bivalent Wuhan + BA.2 formulation.

It is important to note that the final dose of the bivalent formulation contained 20 μg, with 10 μg of the BA.2 Omicron antigen and 10 μg of the Wuhan antigen, whereas the monovalent version of the BA.2 antigen had 20 μg (two times higher). Consequently, if the primary neutralizing effect against the more divergent BQ.1 strain was attributed to the BA.2 Omicron antigen rather than the Wuhan antigen, then maintaining the same dosage of this antigen in both the bivalent and monovalent groups (20 μg) could produce comparable outcomes. Nevertheless, our observations did not reveal any synergistic impact from combining two different antigens in a bivalent formulation in neutralizing the distinct SARS-CoV-2 strain.

Therefore, monovalent formulations of the vaccine worked well against the same strains, particularly against Wuhan and BA.2 variants. Bivalent formulations that contained the same strain also worked but did not show inferiority in comparison with monovalent variants. In the case of the distinct Omicron strain, BQ.1, there was a tremendous reduction in the neutralizing antibodies in all groups, including monovalent and bivalent formulations of Wuhan-RBD-Fc and BA.2-RBD-Fc vaccine antigens. However, the better effect was achieved with more close strain BA.2 (with usage of antigen in a higher dosage) to BQ.1 than in the case of Wuhan to BQ.1. Therefore, we suppose that usage of the closest strain to the circulating one, as in the monovalent as well as in bivalent formulations, might give the best protection and necessary neutralizing antibodies. If there is one prevalent circulating variant, the bivalent formulation might not give the additional superiority in comparison with the monovalent form.

A major limitation of this study is that it was conducted in mice, which do not have previous exposure or immunity to SARS-CoV-2 strains. This contrasts with the real-world context where a considerable proportion of the population has encountered various SARS-CoV-2 variants. At the same time, the abovementioned different outcomes observed in bivalent vaccine efficacy compared to monovalent vaccines may be influenced by individuals’ prior exposure and existing immunity to diverse SARS-CoV-2 strains, potentially affecting bivalent vaccine effectiveness through “imprinting” effects. Therefore, using mice helped mitigate this “imprinting” effect, but it restricts direct extrapolation of findings to human populations, necessitating comprehensive clinical studies that analyze preexisting immunity. Another limitation of this study is the inclusion of three previous SARS-CoV-2 variants, whereas ongoing virus evolution has led to the emergence of newer strains in circulation.

## 5. Conclusions

This study demonstrated that intramuscular immunization of animals with Betuvax-CoV-2 vaccine formulations induced a strong humoral immune response, with a significant rise in specific antibody titers following the second dose. However, the neutralizing capacity of the vaccine varied depending on the SARS-CoV-2 strain, regardless of whether the vaccine was monovalent or bivalent. These findings suggest that for optimal neutralizing antibody response, the vaccine should preferably be closely matched to the circulating SARS-CoV-2 strain and administered at the appropriate dosage. Additionally, this study found no significant advantage in using a bivalent vaccine over a monovalent one when targeting a single dominant SARS-CoV-2 strain.

## Figures and Tables

**Figure 1 vaccines-12-01200-f001:**
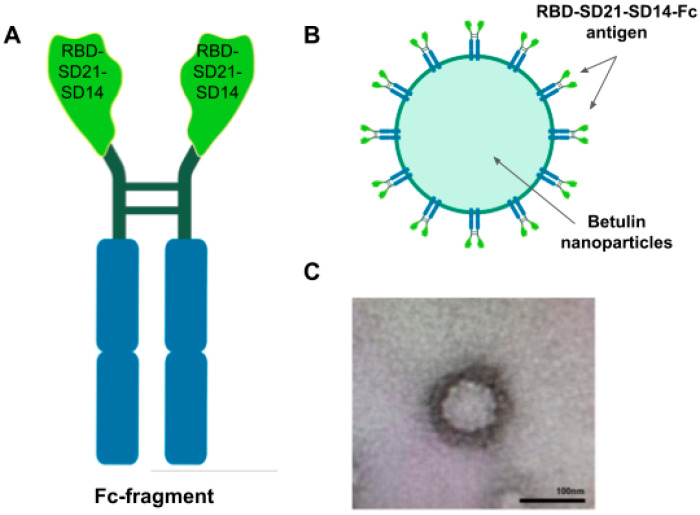
Betuvax-CoV-2 vaccine. Schematic of the RBD-Fc antigen construction (**A**); graphic representation of Betuvax-CoV-2 betulin-based nanoparticle with absorbed antigens (**B**); and transmission electron microscopy of the Betuvax-CoV-2 vaccine, the bar size in 100 nm (**C**).

**Figure 2 vaccines-12-01200-f002:**
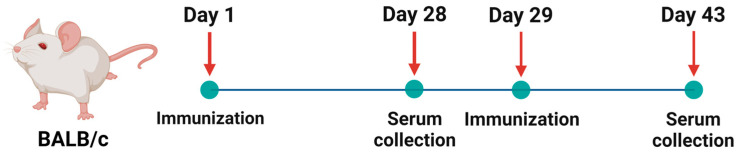
Scheme of BALB/c mice immunization.

**Figure 3 vaccines-12-01200-f003:**
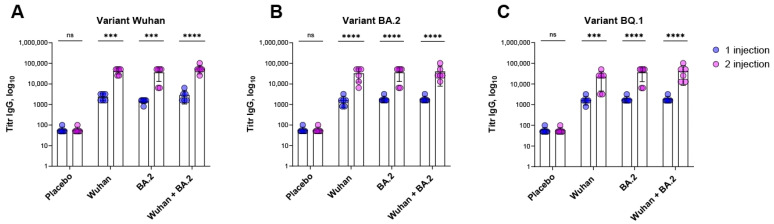
The level of IgG-specific antibodies to the RBD-Fc protein of the SARS-CoV-2 coronavirus variant Wuhan (**A**), BA.2 (**B**), and BQ.1 (**C**) in BALB/c mice after single and double immunization with samples of the candidate vaccine or placebo. The GMT ± standard deviation group is presented. For statistical processing, a two-way ANOVA test with Sidak’s post hoc test was used (**** *p* < 0.0001, *** *p* < 0.001, ns—non-significant).

**Figure 4 vaccines-12-01200-f004:**
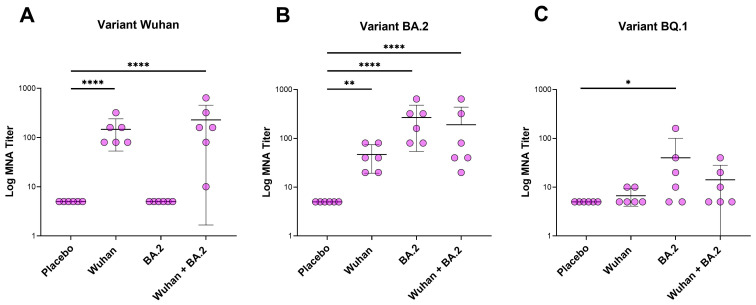
The level of neutralizing antibodies to the SARS-CoV-2 strains Wuhan (**A**), BA.2 (**B**), and BQ.1 (**C**) in BALB/c mice after double immunization with samples of the candidate vaccine. The GMT ± standard deviation group is presented. For statistical processing, a one-way ANOVA test with Dannet’s test was used (**** *p* < 0.0001, ** *p* < 0.01, * *p* < 0.05).

**Table 1 vaccines-12-01200-t001:** Study design and schedule of manipulations during the investigation of the immunological properties of the Betuvax-CoV-2 vaccine containing recombinant antigen RBD-Fc from various variants of the SARS-CoV-2 during intramuscular immunization of mice.

Group №	Preparation	Method of Administration (1 Dosage—0.5 mL)	Number of Mice	Observations and Manipulations by Day			
				Day 1	Day 28	Day 29	Day 43
1	Placebo (Tris–HCl 0.05 M, NaCl 0.15 M)	i/m	6	Immunization 1	Serum collection 1	Immunization 2	Serum collection 2
2	Monovalent vaccine (Wuhan)	i/m	6	Immunization 1	Serum collection 1	Immunization 2	Serum collection 2
3	Monovalent vaccine (BA.2)	i/m	6	Immunization 1	Serum collection 1	Immunization 2	Serum collection 2
4	Bivalent vaccine (Wuhan + BA.2)	i/m	6	Immunization 1	Serum collection 1	Immunization 2	Serum collection 2

## Data Availability

The datasets generated or analyzed during this study are available from the corresponding author on a reasonable request.

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
