# Peer review of "Comparative Analysis of the Neutralizing Capacity of Monovalent and Bivalent Formulations of Betuvax-CoV-2, a Subunit Recombinant COVID-19 Vaccine, Against Various Strains of SARS-CoV-2"

_vaccines, 2024, doi:10.3390/vaccines12101200_

Round 1
Reviewer 1 Report
Comments and Suggestions for Authors
Overall, the manuscript presents an interesting comparison of monovalent and bivalent formulations of the Betuvax-CoV-2 vaccine. However, several revisions are needed to improve the clarity, depth, and comprehensiveness of the work.
The introduction briefly mentions the trend towards bivalent vaccines but does not provide a comprehensive review of existing literature on bivalent COVID-19 vaccines. A discussion of relevant published studies would help situate the current research in the context of existing knowledge.
While the introduction provides a good overview of the COVID-19 pandemic and the importance of vaccines, it lacks specific details on the Betuvax-CoV-2 vaccine and its previous performance. More information on the clinical trial data and safety profile of the monovalent Wuhan-based formulation could strengthen the background.
Animal Model Details: The methods section adequately describes the animal model and ethical considerations but lacks information on the justification for using BALB/c mice for this study. A brief discussion of why this mouse strain was chosen and its relevance to human immune responses would be useful.
The formulations and dosages of the test drugs are described clearly. However, the specifics of the adjuvant used (natural betulin) and its role in enhancing immunogenicity could be expanded upon. Additional details on the stability and storage conditions of the vaccines would also strengthen the methods.
The procedure for immunization and serum collection is well described. However, it would be helpful to include information on the specific techniques used to ensure consistent and accurate blood sampling and serum processing.
The results section lacks specific numerical data and statistical analysis tables, such as tables with mean, standard deviation, and statistical significance values.
The interpretation of the results focuses on the comparison between monovalent and bivalent vaccines but lacks a discussion of the implications for vaccine design and public health policy. A broader discussion of the findings, including their limitations and potential applications, would strengthen the conclusions.
The experimental validation section in this study appears to be relatively thin, and it is recommended to increase other assessments in order to enhance the reliability of this research.
The discussion does not explicitly address the limitations of the study. Including a discussion of potential biases, unmeasured con-founders, and sources of error would enhance the rigor and credibility of the findings.
Comments on the Quality of English LanguageThere are several typos and grammatical errors
Author Response
Dear Reviewer,
Thank you for your valuable suggestions.
Comment:
The introduction briefly mentions the trend towards bivalent vaccines but does not provide a comprehensive review of existing literature on bivalent COVID-19 vaccines. A discussion of relevant published studies would help situate the current research in the context of existing knowledge.
While the introduction provides a good overview of the COVID-19 pandemic and the importance of vaccines, it lacks specific details on the Betuvax-CoV-2 vaccine and its previous performance. More information on the clinical trial data and safety profile of the monovalent Wuhan-based formulation could strengthen the background.
The interpretation of the results focuses on the comparison between monovalent and bivalent vaccines but lacks a discussion of the implications for vaccine design and public health policy. A broader discussion of the findings, including their limitations and potential applications, would strengthen the conclusions.
The experimental validation section in this study appears to be relatively thin, and it is recommended to increase other assessments in order to enhance the reliability of this research.
The discussion does not explicitly address the limitations of the study. Including a discussion of potential biases, unmeasured con-founders, and sources of error would enhance the rigor and credibility of the findings.
Response:
In response, we have expanded both the Introduction and Discussion sections to include a more comprehensive review of existing studies on bivalent COVID-19 vaccines (lines 74-109, 367-388, 409-417, 430-441).
Comment:
Animal Model Details: The methods section adequately describes the animal model and ethical considerations but lacks information on the justification for using BALB/c mice for this study. A brief discussion of why this mouse strain was chosen and its relevance to human immune responses would be useful.
The formulations and dosages of the test drugs are described clearly. However, the specifics of the adjuvant used (natural betulin) and its role in enhancing immunogenicity could be expanded upon. Additional details on the stability and storage conditions of the vaccines would also strengthen the methods.
The procedure for immunization and serum collection is well described. However, it would be helpful to include information on the specific techniques used to ensure consistent and accurate blood sampling and serum processing.
Response:
Additionally, we have revised the Methods section, adding further details regarding our experimental procedures and protocols to enhance clarity and reproducibility (lines 130-191, 200-201).
Comment:
The results section lacks specific numerical data and statistical analysis tables, such as tables with mean, standard deviation, and statistical significance values.
Response:
Regarding numerical data, we have opted to retain the graphical representations as they effectively convey all relevant information and statistical analyses. However, we are happy to provide tables with detailed numerical data upon request.
Thank you once again for your constructive feedback.
Reviewer 2 Report
Comments and Suggestions for Authors
The manuscript entitled “Comparative analysis of the efficacy of monovalent and bivalent formulations of Betuvax-CoV-2, a subunit recombinant COVID-19 vaccine, against various stains of SARS-CoV-2” by Vakhrushova et al. demonstrates a good command of the English language. The study was performed in mice, but the results have relevance to vaccination in humans. Appropriate controls were used, and the experiments seem to have been carefully performed. The results are interesting and make a significant contribution to the field of vaccine studies.
Major Points:
1. Adjust the margins of the pages such that the tables and figures will fit within them. For example, the table on page 3 needs to have the column widths increased to fit the headings, etc.
2. Increase the size of the labels for each of the figures so they can be easily read.
3. In the discussion section, propose some reasons for the greater variation in neutralizing antibody between the various vaccines. Include arguments for why the bivalent vaccine seemed to produce fewer neutralizing antibodies than the monovalent BA.2 vaccine.
4. In humans it has been found that antibodies are generated against conserved sequences within the HA protein of the influenza virus. Could this also be true for the COVID-19 spike protein?
5. In line 318 change “varint” to “variant”.
Author Response
Dear Reviewer,
Thank you for your corrections and insightful feedback.
Comment:
1. Adjust the margins of the pages such that the tables and figures will fit within them. For example, the table on page 3 needs to have the column widths increased to fit the headings, etc.
2. Increase the size of the labels for each of the figures so they can be easily read.
5. In line 318 change “varint” to “variant”.
Response: We have addressed points 1, 2, and 5 as suggested.
Comment:
3. In the discussion section, propose some reasons for the greater variation in neutralizing antibody between the various vaccines. Include arguments for why the bivalent vaccine seemed to produce fewer neutralizing antibodies than the monovalent BA.2 vaccine.
Response:
In the Discussion section, we have provided a broader explanation of the levels of neutralizing antibodies, elaborating on their role and relevance to our findings (lines 367-388, 409-417).
Comment:
4. In humans it has been found that antibodies are generated against conserved sequences within the HA protein of the influenza virus. Could this also be true for the COVID-19 spike protein?
Response:
Regarding your question on influenza and the use of conserved viral regions, we would like to clarify that antibodies targeting these conserved regions are unlikely to be neutralizing and, therefore, may not offer effective protection against the virus.
We appreciate your constructive input and believe these revisions have strengthened the manuscript.
Reviewer 3 Report
Comments and Suggestions for Authors
This manuscript reports a subunit vaccine containing COVID-19 RBD/IgFc fusion protein, "Betuvax-CoV-2", which causes IgG antibody immune response after immunization in mice, and has certain research value for exploring the protective effect of the vaccine;
The content of this manuscript is relatively thin and a research report on a phenomenon, lacking in-depth exploration of the mechanism and requiring further optimization;
This manuscript does not display the structure and sequence of different RBD regions used, and needs to be supplemented;
This manuscript only includes research on mice, which limits its scientific value. It is suggested to supplement the research results on humans;
This manuscript needs to further highlight the characteristics and advantages of the Betuvax-CoV-2 vaccine;
This manuscript needs to discuss the limitations of conducting this study in mice and their impact on the conclusions.
Author Response
Dear Reviewer,
Thank you very much for your consideration and valuable feedback.
Comment: The content of this manuscript is relatively thin and a research report on a phenomenon, lacking in-depth exploration of the mechanism and requiring further optimization;
This manuscript only includes research on mice, which limits its scientific value. It is suggested to supplement the research results on humans;
Response: Indeed, this study complements our previous extensive work on the structural description of the vaccine (https://doi.org/10.3390/vaccines10010069), as well as its preclinical and clinical results in humans in the monovalent form (https://doi.org/10.3390/vaccines10081290, https://doi.org/10.3390/vaccines11020326). However, this particular study focuses on comparing the bivalent and monovalent formulations of the subunit recombinant vaccine Betuvax-CoV-2.
Comment: This manuscript does not display the structure and sequence of different RBD regions used, and needs to be supplemented;
This manuscript needs to further highlight the characteristics and advantages of the Betuvax-CoV-2 vaccine;
Response: To avoid redundancy with previously published findings, we have provided relevant references within the article. In response to your suggestion, we have added Figure 1 (page 3), illustrating the structure of the antigen and vaccine, along with its description and key advantages (lines 76-109).
Comment: This manuscript needs to discuss the limitations of conducting this study in mice and their impact on the conclusions.
Response: Additionally, we have expanded the Discussion with the limitations of this study (lines 430-441).
Thank you again for your insightful comments.
Reviewer 4 Report
Comments and Suggestions for Authors
The article written by Anna Vakhrusheva on "comparative analysis of the efficacy of monovalent and bivalent formulations of Betuvax-CoV-2, a subunit recombinant COVID-19 vaccine, against various strains of SARS-CoV2" tested monovalent and bivalent vaccine formulation in mice model.
Though it is well-written including methods and the data interpretation are valid, it didn't provide any novel scientific information/methodology. The results shown by them were already published by many researchers years ago making the current article less scientific interest for the readers. For example, the fact that spike-RBD is highly immunogenic and induces robust antibody responses were already reported in mice (https://www.nature.com/articles/s41586-020-2599-8).
2. Better neutralizing antibody responses are induced only after the 2nd dose is also reported earlier (https://bmcmedicine.biomedcentral.com/articles/10.1186/s12916-021-02090-6)
3. The current study/report could be improved by in-depth analysis like T cell responses or protection after a challenge that would provide additional information and makes the article more interesting for the reader.
Author Response
Dear Reviewer,
Thank you very much for your valuable suggestions.
Comment: Though it is well-written including methods and the data interpretation are valid, it didn't provide any novel scientific information/methodology. The results shown by them were already published by many researchers years ago making the current article less scientific interest for the readers. For example, the fact that spike-RBD is highly immunogenic and induces robust antibody responses were already reported in mice (https://www.nature.com/articles/s41586-020-2599-8).
Better neutralizing antibody responses are induced only after the 2nd dose is also reported earlier (https://bmcmedicine.biomedcentral.com/articles/10.1186/s12916-021-02090-6)
The current study/report could be improved by in-depth analysis like T cell responses or protection after a challenge that would provide additional information and makes the article more interesting for the reader.
Response:
The novelty of this study lies in the comparison between the bivalent and monovalent formulations of the subunit recombinant vaccine Betuvax-CoV-2.
In our previous publications (https://doi.org/10.3390/vaccines10010069, https://doi.org/10.3390/vaccines10081290, https://doi.org/10.3390/vaccines11020326), we demonstrated the T-cell responses, neutralizing effects, and protective efficacy of the monovalent form of Betuvax-CoV-2, along with its preclinical and clinical evaluations.
In this work, we aimed to determine whether the bivalent formulation enhances vaccine neutralization capacity. Given that most studies focus on bivalent mRNA vaccines, we believe this investigation into bivalent subunit recombinant vaccines, particularly Betuvax-CoV-2, significantly broadens the scope of this field.
Therefore, we have updated the manuscript and added some details to highlight the focus of the study (lines 74-109, 367-388, 409-417, 430-441).
Thank you again for your insightful feedback.
Round 2
Reviewer 1 Report
Comments and Suggestions for Authors
The content of the manuscript has been significantly enriched and improved compared with the previous version. However, there are still several key issues that have not been resolved.
1. Lack of Source Information for Key Experimental Materials:
The manuscript lacks information on the sources and purity of key experimental materials used, such as the recombinant RBD-Fc proteins and the live SARS-CoV-2 strains employed in ELISA and neutralization assays. Providing details on the manufacturer, catalog numbers, and any purification or quality control procedures used is essential for reproducibility and verification of the results.
2. Insufficient Discussion of Limitations:
The authors briefly mention that the study was conducted in mice, which may not fully reflect the immune response in humans. However, a more comprehensive discussion of other potential limitations, such as the impact of prior exposure or immunity to SARS-CoV-2 strains in humans, is warranted. This would help readers contextualize the results and identify areas for future research.
3. Regarding the response, a targeted reply should be made in accordance with the standard reply method for the proposed revision suggestions, rather than simply marking the approximate location in the article after the change.
Comments on the Quality of English LanguageMinor editing of English language required.
Author Response
Dear Reviewer,
Thank you very much for your consideration and valuable feedback.
Comment 1: Lack of Source Information for Key Experimental Materials:
The manuscript lacks information on the sources and purity of key experimental materials used, such as the recombinant RBD-Fc proteins and the live SARS-CoV-2 strains employed in ELISA and neutralization assays. Providing details on the manufacturer, catalog numbers, and any purification or quality control procedures used is essential for reproducibility and verification of the results.
Response 1: We have clarified the SARS-CoV-2 strains used in the study (p. 6, lines 232-235). An attachment with the characteristic of the recombinant RBD-Fc protein is provided in the Supplementary file.
Comment 2: Insufficient Discussion of Limitations:
The authors briefly mention that the study was conducted in mice, which may not fully reflect the immune response in humans. However, a more comprehensive discussion of other potential limitations, such as the impact of prior exposure or immunity to SARS-CoV-2 strains in humans, is warranted. This would help readers contextualize the results and identify areas for future research.
Response 2: Thank you for pointing this out; we agree with this notion and have added this description in the Discussion section (p. 11-12, lines 430-441)
Comment 3: Regarding the response, a targeted reply should be made in accordance with the standard reply method for the proposed revision suggestions, rather than simply marking the approximate location in the article after the change.
Response 3: We have updated the answers based on your previous revision as follows:
The introduction briefly mentions the trend towards bivalent vaccines but does not provide a comprehensive review of existing literature on bivalent COVID-19 vaccines. A discussion of relevant published studies would help situate the current research in the context of existing knowledge. We have included this description in the Discussion part (p. 10-11, lines 367-388).
While the introduction provides a good overview of the COVID-19 pandemic and the importance of vaccines, it lacks specific details on the Betuvax-CoV-2 vaccine and its previous performance. More information on the clinical trial data and safety profile of the monovalent Wuhan-based formulation could strengthen the background. This part has been added to the Introduction section (p. 3, lines 93-100).
Animal Model Details: The methods section adequately describes the animal model and ethical considerations but lacks information on the justification for using BALB/c mice for this study. A brief discussion of why this mouse strain was chosen and its relevance to human immune responses would be useful. The information has been added to the Methods section (p. 4, lines 130-132).
The formulations and dosages of the test drugs are described clearly. However, the specifics of the adjuvant used (natural betulin) and its role in enhancing immunogenicity could be expanded upon. Additional details on the stability and storage conditions of the vaccines would also strengthen the methods. Details about betulin have been included in the Introduction (p. 2-3, Fig. 1, lines 75-77) and further elaborated in a previously cited article (DOI:10.3390/vaccines10010069). Storage conditions were detailed in the Methods section (p. 5, lines 190-191).
The procedure for immunization and serum collection is well described. However, it would be helpful to include information on the specific techniques used to ensure consistent and accurate blood sampling and serum processing. This information has been added to the Methods part (p. 5-6, lines 199-205).
The results section lacks specific numerical data and statistical analysis tables, such as tables with mean, standard deviation, and statistical significance values. Statistics have been indicated in the description under Figures 3 and 4 (p. 8-9). Tables with numerical data can be provided upon request to avoid overloading the visual perception of information.
The interpretation of the results focuses on the comparison between monovalent and bivalent vaccines but lacks a discussion of the implications for vaccine design and public health policy. A broader discussion of the findings, including their limitations and potential applications, would strengthen the conclusions. Additional information has been included in the Discussion (p. 11, lines 409-429).
The experimental validation section in this study appears to be relatively thin, and it is recommended to increase other assessments in order to enhance the reliability of this research. All necessary statistical analyses and comprehensive assessments related to this data are presented on p. 7, lines 254-262.
The discussion does not explicitly address the limitations of the study. Including a discussion of potential biases, unmeasured con-founders, and sources of error would enhance the rigor and credibility of the findings. The Discussion part has been expanded to include the limitations of the study (p.11-12, lines 430-441).

Reviewer 4 Report
Comments and Suggestions for Authors
The author has made a few changes adding additional details to the manuscript making it a scientifically improved version. Also, answered about the novelty of their research.
Author Response
Dear Reviewer,
Thank you for your valuable feedback, which has helped us improve the manuscript.
Round 3
Reviewer 1 Report
Comments and Suggestions for Authors
Most of comments have been sufficiently addressed in this round.